# Safety of Administering Live Vaccines during Pregnancy: A Systematic Review and Meta-Analysis of Pregnancy Outcomes

**DOI:** 10.3390/vaccines8010124

**Published:** 2020-03-11

**Authors:** Almudena Laris-González, Daniel Bernal-Serrano, Alexander Jarde, Beate Kampmann

**Affiliations:** 1Hospital Infantil de México Federico Gómez, Mexico 06720, Mexico; almu_laris@hotmail.com; 2London School of Hygiene and Tropical Medicine (alumni), London WC1E 7HT, UK; Dbernal@tec.mx; 3Compañeros en Salud—Partners in Health México, Mexico 11800, Mexico; 4Instituto Tecnológico y de Estudios Superiores de Monterrey, Mexico 14380, Mexico; 5Disease Control Elimination Theme, MRC Unit The Gambia at the London School of Hygiene and Tropical Medicine, Banjul P.O. Box 273, Gambia; ajarde@mrc.gm; 6The Vaccine Centre, Faculty of Infectious and Tropical Diseases, London School of Hygiene and Tropical Medicine, London WC1E 7HT, UK; 7Vaccines & Immunity Theme, MRC Unit The Gambia at the London School of Hygiene and Tropical Medicine, Banjul P.O. Box 273, Gambia

**Keywords:** live attenuated vaccines, safety, pregnant women, pregnancy outcomes

## Abstract

Live-attenuated vaccines (LAV) are currently contraindicated during pregnancy, given uncertain safety records for the mother–infant pair. LAV might, however, play an important role to protect them against serious emerging diseases, such as Ebola and Lassa fever. For this systematic review we searched relevant databases to identify studies published up to November 2019. Controlled observational studies reporting pregnancy outcomes after maternal immunization with LAV were included. The ROBINS-I tool was used to assess risk of bias. Pooled odds ratios (OR) were obtained under a random-effects model. Of 2831 studies identified, fifteen fulfilled inclusion criteria. Smallpox, rubella, poliovirus, yellow fever and dengue vaccines were assessed in these studies. No association was found between vaccination and miscarriage (OR 0.98, 95% CI 0.87–1.10), stillbirth (OR 1.04, 95% CI 0.74–1.48), malformations (OR 1.09, 95% CI 0.98–1.21), prematurity (OR 0.99, 95% CI 0.90–1.08) or neonatal death (OR 1.06, 95% CI 0.68–1.65) overall. However, increased odds of malformations (OR 1.24; 95% CI 1.03–1.49) and miscarriage after first trimester immunization (OR 4.82; 95% CI 2.38–9.77) was found for smallpox vaccine. Thus, we did not find evidence of harm related to LAV other than smallpox with regards to pregnancy outcomes, but quality of evidence was very low. Overall risks appear to be small and have to be balanced against potential benefits for the mother-infant pair.

## 1. Introduction

Pregnancy induces immune modulation that renders women more susceptible to severe manifestations of infectious diseases, some of which can potentially be prevented by vaccination, e.g., influenza. In addition to protecting the mother, vaccination during pregnancy also serves as a vehicle to protect the unborn or newborn child via high-titer transplacental antibodies which can safeguard infants until active immunity through childhood vaccination has been established [1].

Scientific evidence supporting the effectiveness of certain inactivated vaccines in pregnancy is growing. Maternal immunization with tetanus, inactivated influenza and pertussis containing vaccines is currently recommended in many countries based on extensive data supporting their safety, and proven benefits for the mother–infant pair [1,2,3].

Live viral vaccines (LAV), on the other hand, can replicate in the host and are potentially capable of causing viremia [4,5,6], conveying the risk of transplacental transmission to the developing fetus with potential adverse consequences on pregnancy outcomes. Safety of vaccination during the first trimester is of particular concern. During this period, women are more likely to be inadvertently vaccinated in the context of mass campaigns, as they might not yet be aware of their pregnancies.

The theoretical risks associated with LAV have led to the exclusion of pregnant women from clinical trials, hindering the development of high-quality evidence regarding potential benefits and harms in this group. Despite the vast employment of LAV in diverse settings, current evidence comes mainly from observational studies and national teratology registries [7].

The extensive use of smallpox vaccine in the 20th century brought about dozens of reported cases of fetal vaccinia, resulting in stillbirth or neonatal death. The risk of fetal vaccinia has been estimated at one per 10,000 to 100,000 immunizations during pregnancy [8,9].

With regards to rubella vaccine, there is evidence of vertical transmission and subclinical fetal infection, as demonstrated by immunoglobulin M (IgM) titers in cord blood of exposed infants [10,11,12], and more recently by molecular methods [13].

Oral poliovirus vaccine (OPV) has been widely used in mass immunization campaigns targeting children and adults, including pregnant women. Transplacental infection with wild-type poliovirus has been documented [14], and a fatal case of spinal cord damage after maternal immunization was described [15]. However, fetal infection with OPV has not been definitely proven. Similarly, yellow fever (YF) vaccine is capable of causing viremia but its potential for causing fetal infection is uncertain [16,17].

Major health benefits may be lost if maternal immunization is withheld on the basis of unproven safety concerns, especially in the case of diseases that disproportionately affect pregnant women and their offspring, and in outbreak scenarios. In recent years, the epidemics of Ebola and Zika have led to an enhanced interest in the role of LAV in pregnant women and the ethical issues surrounding their exclusion in clinical trials in such scenarios [18]. Furthermore, it is important to improve our knowledge of the safety of vaccines that might be used in mass campaigns where pregnant women may be inadvertently vaccinated, in order to inform future research and health policy.

Given the increasing likelihood that LAV could be used to combat emerging infectious disease such as Ebola, Lassa and others, we aimed to assess the body of evidence evaluating the safety of maternal immunization with LAV, and to identify the main areas of uncertainty where new studies could be beneficial.

## 2. Materials and Methods

For this systematic review and meta-analysis we searched MEDLINE, EMBASE, Cochrane Central Register of Controlled Trials (CENTRAL), Web of Science, Global Health, ClinicalTrials.gov, as well as Open Grey and MedNar for published and unpublished studies from the inception of each database. The search strategy was constructed in collaboration with a medical librarian (Appendix A). The reference list of all included articles was searched for additional studies. The initial search was conducted on June 2019 and updated on 20 November 2019.

Primary studies of experimental and epidemiological study designs were deemed eligible for inclusion if adverse pregnancy outcomes (i.e., miscarriage, stillbirth, neonatal death, prematurity, low birth-weight, birth defects and congenital infection) were assessed in mother-infant pairs exposed to immunization with one or more LAV during pregnancy or in the three months before conception, and compared to mother-infant dyads with no exposure during or immediately before gestation. Narrative reviews, case series, passive surveillance studies with no clear denominator, observational studies with no control group, and articles with unavailable full text were excluded from the review. No language or geographic restrictions were applied.

The search results from each database were exported to Mendeley 1.19.4 and deduplicated. Two reviewers (A.L. and D.B.) independently screened titles and abstracts, reviewed full texts of potentially relevant articles against inclusion criteria, and extracted data into a previously designed data collection form. Extracted data included study design, funding, country and setting, characteristics of the exposed and control groups, potential confounders, number of participants, vaccine(s) administered, trimester of pregnancy at vaccination, outcome definition and time points, number of events and sample size for all prespecified outcomes, crude and adjusted estimates of effect and variables included in adjusted analyses.

Included studies were independently evaluated by both reviewers using the ROBINS-I tool (“Risk Of Bias In Non-randomized Studies—of Interventions”) [19]. The GRADE (Grading of Recommendations Assessment, Development and Evaluation) approach was employed to assess overall quality of evidence for the prespecified outcomes. Any discrepancies were solved by discussion and consensus, consulting a third and fourth reviewer if necessary.

### Data Analysis

A narrative synthesis was used to summarize the characteristics, results and internal validity of included studies, by vaccine. Meta-analyses were carried out using Review Manager software 5.3 (Cochrane Collaboration, Oxford, UK) for outcomes for which studies were deemed comparable. Pooled estimates of effect were obtained using the generic inverse-variance method under a random effects model, and presented as odds ratios (OR) and their 95% confidence intervals (CI). Fixed zero-cell correction was used for studies with no events in one or more groups. The degree of variability between studies was assessed using the I^2^ statistic and heterogeneity was evaluated according to the Cochrane Handbook guidelines.

Given the variety of vaccine types that we encountered and their potentially different effect on pregnancy outcomes, we decided to do subgroup analyses by vaccine type, which had not been pre-specified in the protocol. Sensitivity analyses were conducted excluding studies at critical risk of bias. Given the greater susceptibility of the embryo to potential harmful effects of vaccines, we also decided to perform sensitivity analyses for the association of vaccination in the first trimester of pregnancy with miscarriage and congenital anomalies.

In addition to the studies included in the meta-analyses according to prespecified criteria, cohorts with no comparison group were summarized to obtain further safety data, especially regarding rare outcomes. The risk of congenital rubella syndrome (CRS) was calculated from studies of rubella vaccination, including controlled and uncontrolled cohorts. The upper limit for the risk of CRS was based on two-tailed 95% CI from the Poisson distribution.

There was no funding source for this study. The study protocol was registered with PROSPERO, number CRD42019138360.

## 3. Results

The search strategy identified 2831 articles following the removal of duplicates, of which 15 studies were included in the main body of this review: 12 cohorts (six retrospective, five prospective and one historically-controlled), two case-controls, and one secondary data analysis from clinical trials (Figure 1).

An overview of the key characteristics of included studies is provided Table 1. All retrievable articles were published in English and focused on populations from high and middle income countries (see Appendix B). Sample size varies considerably, from a few dozens to over 9000 participants per group. Only one was industry-funded [20]. Included studies reported on outcomes after immunization with one of the following vaccines: smallpox (eight studies), rubella (three), OPV (two), YF (one), and dengue (one).

### 3.1. Meta-Analysis of Pregnancy Outcomes

Meta-analysis was carried out for the outcomes of miscarriage, stillbirth, congenital anomalies, prematurity, and neonatal death (See Table 2). Data on congenital infection and LBW was considered inadequate for meta-analysis given the scarcity of studies reporting these outcomes, and the heterogeneity in the methods of outcome assessment for congenital infection.

#### 3.1.1. Miscarriage

Four studies on smallpox vaccine, two on rubella, and one each on OPV, YF, and dengue vaccine reported on miscarriage. Meta-analysis showed no association between immunization and spontaneous abortion overall and for each type of vaccine (pooled OR: 0.98; 95% CI 0.87–1.10). No statistical heterogeneity was detected across studies (I^2^ = 0%) (Figure 2a).

The analysis for exposure in the first trimester of pregnancy revealed a pooled OR of 2·66 (95% CI 0.73–9.64). However, the subgroup analysis showed a strong association between smallpox vaccination and miscarriage (OR 4.82, 95% CI 2.38–9.77, *p* < 0·0001) as shown in Figure 2b.

Quality of evidence was rated as very low, given a critical risk of selection bias (Appendix A). When considering exposure to vaccination in the first trimester, downgrading was also granted for imprecision. Funnel plot revealed no evidence of publication bias (Appendix A).

#### 3.1.2. Stillbirth

No association was found between vaccination and stillbirth (pooled OR: 1.04; 95% CI 0.74–1.48) in the pooled analysis of nine studies, including six on smallpox, and one each on rubella, OPV, and dengue vaccine (Figure 3). There was moderate statistical heterogeneity between studies (I^2^ = 45%, *p* = 0.07) and quality of evidence was rated as very low due to serious risk of bias (Appendix A). Funnel plot revealed no evidence of publication bias (Appendix A).

#### 3.1.3. Congenital Anomalies

Eight studies on smallpox vaccine, two on rubella, and two on OPV contributed data for the meta-analysis of congenital anomalies, revealing no evidence of an association with maternal immunization (pooled OR: 1.09; 95% CI 0.98–1.21); this also held true when considering only pregnancies exposed to vaccination in the first trimester (pooled OR: 1.22; 95% CI 0.87–1.72), as depicted in Figure 4a,b. Nonetheless, the subgroup analysis revealed an increase in the odds of congenital anomalies after smallpox vaccination (OR: 1.24; 95% CI 1.03–1.49) and a tendency towards an association with rubella vaccine, albeit with a very wide confidence interval (OR: 2.8; 95% CI 0.65–12.04).

No statistical heterogeneity was detected across studies for the overall effect of immunization on congenital defects (I^2^ = 0%), but there was substantial heterogeneity (I^2^ = 77.4%) between different vaccines when considering only first trimester immunization (*p* = 0·01 for subgroup differences, Figure 4B). Quality of evidence was rated as very low, due to serious risk of bias (Appendix A) and suspicion of publication bias (Appendix A).

#### 3.1.4. Preterm Birth

Five studies were pooled in the meta-analysis for preterm birth, including two on smallpox, two on rubella vaccine, and one on OPV. No association was found between vaccination and prematurity (pooled OR: 0.99; 95% CI 0.90–1.08) overall and for each type of vaccine (Appendix A). No statistical heterogeneity was detected across studies (I^2^ = 0%), but there was moderate heterogeneity between subgroups (I^2^ = 42·7%). Quality of evidence was rated as very low as a result of serious risk of bias (Appendix A).

#### 3.1.5. Neonatal Death

Three studies on smallpox vaccine, one on rubella, and one on OPV contributed to the meta-analysis of neonatal death, showing no association with vaccination (pooled OR: 1.06; 95% CI 0.68–1.65), as depicted in Appendix A. No heterogeneity between estimates of effect was observed (I^2^ = 0%). Quality of evidence was rated as very low due to serious risk of bias (Appendix A) and imprecision.

#### 3.1.6. Sensitivity Analysis Excluding Studies at Critical Risk of Bias

Sensitivity analysis excluding four out of nine studies at critical risk of bias for the outcome of miscarriage revealed no association with vaccination (pooled OR: 0.97, 95% CI 0.86–1.09), consistent with the findings of the analysis including all studies. In the case of miscarriage after first trimester vaccination, all studies were at critical risk of bias and thus sensitivity analysis could not be conducted. For the rest of outcomes there were no studies at critical risk of bias.

### 3.2. Uncontrolled Cohorts

Additionally, twenty-three uncontrolled cohorts [8,10,11,12,13,22,32,38,39,40,41,42,43,44,45,46,47,48,49,50,51,52,53,54,55,56,57,58,59,60,61] describing pregnancy outcomes after maternal vaccination were retrieved in the literature search. No cases of fetal vaccinia were found among 643 women vaccinated against smallpox [8,58,59], and rates of adverse pregnancy outcomes were within expected limits, except for a high frequency of stillbirth among those vaccinated in the first trimester [59] (Appendix A).

Two controlled and twelve uncontrolled studies reported on the prevalence of CRS after maternal immunization [10,11,12,13,22,32,38,39,40,41,42,43,60,61], as shown in Appendix A. No cases were detected among 3918 infants, including 2303 born to women susceptible to rubella before vaccination. The upper bound of the 95% CI for the risk of CRS is 0.09% for all infants, and 0.16% for those born to susceptible women. These cohorts included participants receiving the Cendehill, HPV-77 and RA 27/3 viral strains, which might differ in their fetotropic potential [10].

No signs of unexpected adverse pregnancy outcomes were found in four uncontrolled cohorts of YF vaccine [16,17,62,63,64], except for an increase in minor dysmorphisms in a Brazilian cohort, which was probably explained by detection bias [62]. Only one case of probable intrauterine infection was detected among 422 infants tested for YF virus IgM (Appendix A).

## 4. Discussion

According to the author´s knowledge, this is the first systematic review on the safety of the administration of LAV during pregnancy. Available evidence comes mainly from cohorts of women vaccinated against smallpox and, to a lesser extent, from observational studies regarding rubella, OPV, YF, and dengue vaccines. With the exception of smallpox vaccine, maternal immunization with these other vaccines shows no evidence of impact on pregnancy outcomes, but available evidence is of very poor quality.

Smallpox vaccine is the only LAV clearly implicated in adverse pregnancy outcomes, given the documented cases of fetal vaccinia, but, according to the available evidence, the risk of this severe complication appears to be low. In the present review, an increase in the odds of birth defects was detected and an association between vaccination in the first trimester and miscarriage was also apparent. However, the studies evaluating this outcome were at critical risk of bias because women were identified at prenatal clinics or hospitals, and early pregnancy losses were likely to be missed, which could have decreased the effect estimate. Furthermore, some studies included all vaccinated women, while others assessed only those considered as “successfully vaccinated”, i.e., subjects who developed a cutaneous reaction. The proportion of women who underwent primary vaccination as opposed to revaccination during pregnancy varied widely between studies, and preexisting immunity could result in different responses to smallpox vaccine (see Appendix A).

The largest studies included in this review were two population-based cohorts, comprising nearly 30,000 women, exploring the safety of OPV in the context of mass vaccination campaigns and showed no increased risk of adverse pregnancy outcomes [27,35]. In neither of these studies was exposure to vaccination determined at the individual level. However, mass campaign coverage was very high in both cases, increasing the confidence in their findings.

Despite the clear teratogenic action of wild-type rubella virus and the evidence of intrauterine infection in infants of women vaccinated during pregnancy, no cases of CRS have been documented in large cohorts of women inadvertently vaccinated during pregnancy or before conception in diverse settings and with different vaccine strains, which might have different fetotropic and teratogenic potential [10]. Maternal immunization with rubella vaccine does not appear to increase the risk of adverse pregnancy outcomes according to the available evidence from populations of women with varying degrees of previous immunity to rubella. However, the diversity of methods used to ascertain susceptibility and congenital infection, including hemagglutination-inhibition, IgM and IgG assays, IgG avidity tests, and viral isolation, could result in variable diagnostic accuracy.

Even though a total absence of risk cannot be proven, the likelihood of CRS is less than one per 1000 exposed pregnancies, according to available evidence [10,11,12,13,22,32,38,39,40,41,42,60,61]. Therefore, women inadvertently vaccinated during pregnancy should be reassured and therapeutic abortion would not be justified on the basis of a teratogenic potential of the vaccine.

The evidence regarding other LAV is even more scarce. In the case of YF vaccine, the only controlled study suggested an increased odds of miscarriage among vaccinated women, but the evidence for the association was weak. The mass vaccination campaign that provided the setting for this study was conducted after a dengue outbreak where organophosphate insecticides had also been used for vector control. Although reported exposure to organophosphates was similar between groups, measurement was imprecise, and residual confounding cannot be ruled out.

Other pregnancy outcomes after YF vaccination have been described only in uncontrolled studies. The largest of these cohorts followed 441 women inadvertently vaccinated in a mass campaign in Brazil [16,62]; the frequency of stillbirths (0.7%) and preterm birth (7.8%) were similar or lower than regional rates as reported by the authors. Major birth defects were found in 3.3% of 304 newborns. Minor dysmorphisms were detected in 62 infants, a frequency greater than expected, but this was attributed to evaluation bias. IgM antibodies were undetectable in blood samples from 341 infants, but antibody persistence was detected in one of 37 children assessed after two years of age, raising the possibility of intrauterine infection.

The only study analyzing the safety of maternal immunization with dengue vaccine suggested a possible increase in the frequency of stillbirths [20], but there were only two events in the exposed group, both in adolescent mothers. This association, however, should be furthered explored.

No controlled studies fulfilling the inclusion criteria were found for the live-attenuated influenza vaccine (LAIV). Similarly, no studies on varicella-zoster virus (VZV) vaccine fulfilled inclusion criteria, but data on 893 women accidentally vaccinated during or shortly before conception is available from the Merck/CDC Pregnancy Registry [65,66]. The prevalence of congenital anomalies (2.1%) was similar to that reported in the U.S. with no specific pattern of birth defects [67]. No cases of congenital varicella syndrome were detected among 810 infants.

The live-attenuated viral vectored vaccine against Ebola (rVSV-ZEBOV) is currently being administered to pregnant women as part of the ring vaccination strategy in Democratic Republic of Congo [68]. This will provide an opportunity to evaluate pregnancy outcomes to better inform decision-making in future outbreak scenarios.

Although the results presented in this manuscript are mostly reassuring, there are several limitations which preclude drawing final conclusions regarding the safety of maternal immunization with LAV. Besides the observational nature of the available evidence, most studies did not follow participants from the time of vaccination, leading to incomplete ascertainment of early pregnancy loses, which might include cases of congenital infection and malformations. Variable diagnostic criteria, ancillary methods and follow-up for ascertainment of birth defects can result in varying degrees of underreporting. The heterogeneity in study designs, settings, definitions and data collection methods leads to challenges in the comparison of different studies. On the other hand, the wide variety of countries and settings increases the generalizability to different populations. Furthermore, each LAV might have a different effect on pregnancy outcomes, which might be obscured by obtaining pooled effects. However, this was addressed by subgroup analyses which revealed an increased odds of miscarriage and congenital anomalies only after smallpox vaccination.

All but one of the included studies were considered at serious risk of bias due to confounding for all outcomes, since most of them did not measure potential confounding factors or did not control for them. Only the studies by Nishioka and Ryan provided an effect estimate adjusted for confounding [33,36]. However, confounding domains were not measured validly in the first case, and some important confounders were not controlled for in the second. Bar-Oz et al. controlled for confounding by matching, but no information was provided on the methods used to measure these potential confounders [22].

Four out of nine studies were considered at critical risk of selection bias for the outcome of spontaneous abortion [24,25,30,31], because participants were selected among women attending participating hospitals or clinics, thus missing those receiving care in different facilities and miscarriages occurring before onset of prenatal care.

The findings from this literature review cannot be directly extended to other LAV given the variable teratogenic potential of different vaccines. The use of modern technologies, such as viral vectored vaccines and novel adjuvants, might also modify the safety profile of live vaccines available in the near future. Furthermore, the power to detect a modest increase in the frequency of specific types of malformations is low, so the association of maternal vaccination with birth defects cannot be ruled out.

The development of new LAV that may target women of childbearing age, such as the viral vectored candidates for the prevention of Zika [69,70], and the use of the existing ones in mass vaccination campaigns or in response to outbreaks will need to be coupled with enhanced surveillance of pregnant women and their infants, as well as standardized case definitions to better characterize the impact of vaccination on the mother-offspring dyad [71]. Additionally, more accurate baseline information on maternal and fetal outcomes in low and middle income countries (LMIC) is desirable to make comparisons with appropriate local rates [71].

Furthermore, other factors need to be taken into account when considering maternal immunization, including implementation challenges and the need for integration with existing prenatal care services, as well as the potential interference of maternal antibodies with the development of infant humoral immune responses to vaccines in the first few months of life. This interference has been observed for several vaccines but its clinical significance is still uncertain and might differ depending on the specific antigen [72,73].

In summary, the risk of adverse pregnancy outcomes after maternal immunization with LAV appears to be small. This information can support decision-makers in the planning of vaccination campaigns that target women in the reproductive age, and may provide reassurance to healthcare workers taking care of women inadvertently immunized with LAV during pregnancy.

Our findings may also inform pressing decisions regarding vaccination of pregnant women during outbreaks. In this setting, benefits might exceed potential harms when the disease manifestations are severe for pregnant women and/or their offspring and the probability of exposure is high, as is the case with Ebola and Lassa Fever [74,75].

Inclusion of pregnant women in vaccine trials under conditions of enhanced safety monitoring and appropriate follow-up of mothers and infants has the potential to expand the benefits of immunization to these populations. The views of pregnant women themselves need to be taken into account in research design and policy making. Qualitative research on vaccine confidence [18,76] and willingness to participate in clinical trials, conducted in different settings and geographic locations, can help improve acceptance. We believe that pregnant women ought to be protected through research, not from research, in order to move forward in the promotion of health equity.

## Figures and Tables

**Figure 1 vaccines-08-00124-f001:**
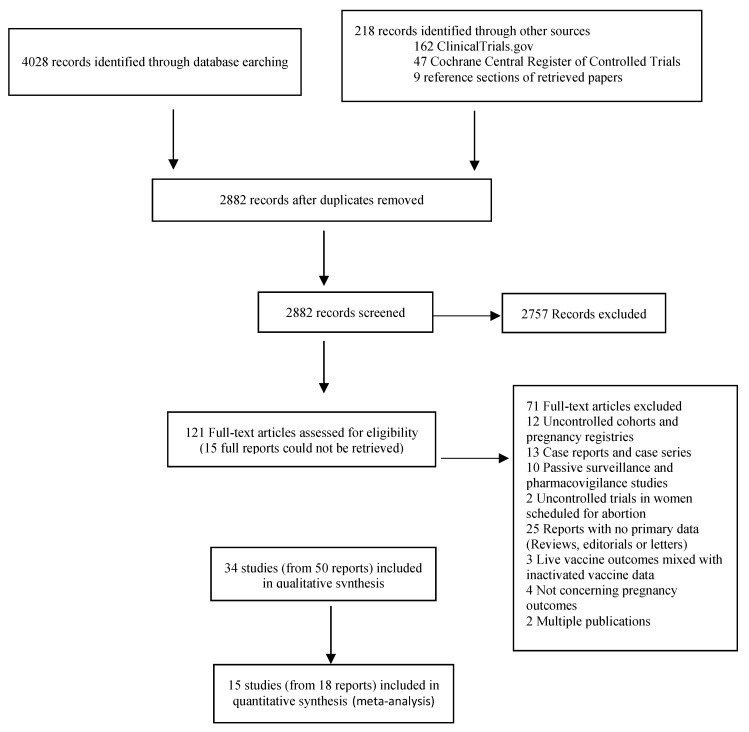
Flowchart showing the study selection process for the qualitative and quantitative synthesis.

**Figure 2 vaccines-08-00124-f002:**
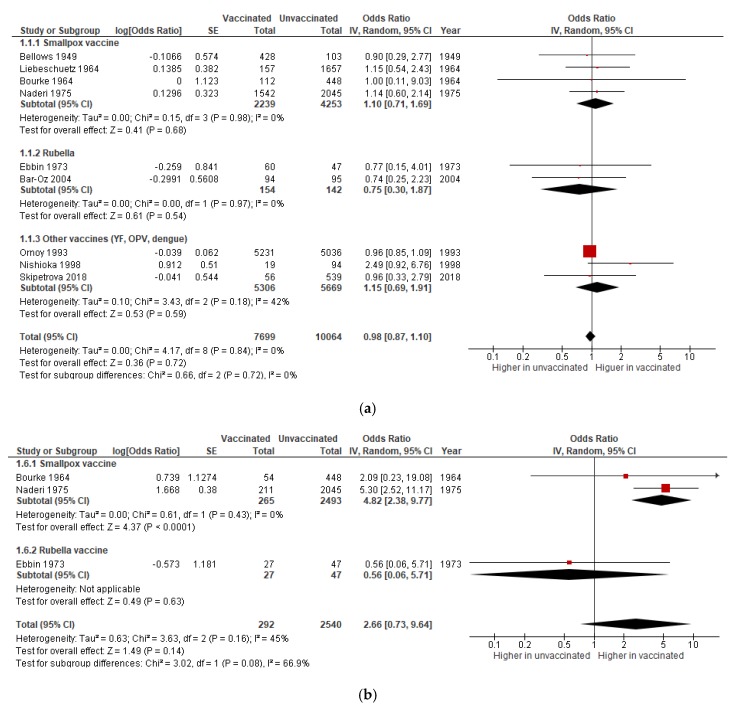
Meta-analysis of the association between maternal immunization and miscarriage. Forest plots showing the effect of vaccination during pregnancy on the odds of miscarriage in all trimesters (**a**) and in the first trimester (**b**), subgrouped by vaccine.

**Figure 3 vaccines-08-00124-f003:**
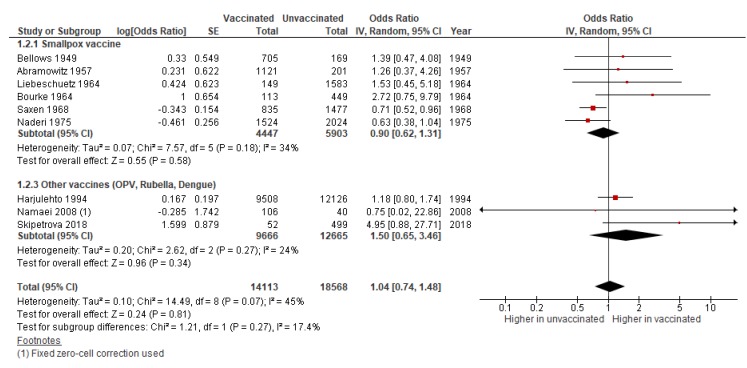
Meta-analysis of the association between maternal immunization and stillbirth. Forest plot showing the effect of immunization during pregnancy on the odds of stillbirth, subgrouped by vaccine.

**Figure 4 vaccines-08-00124-f004:**
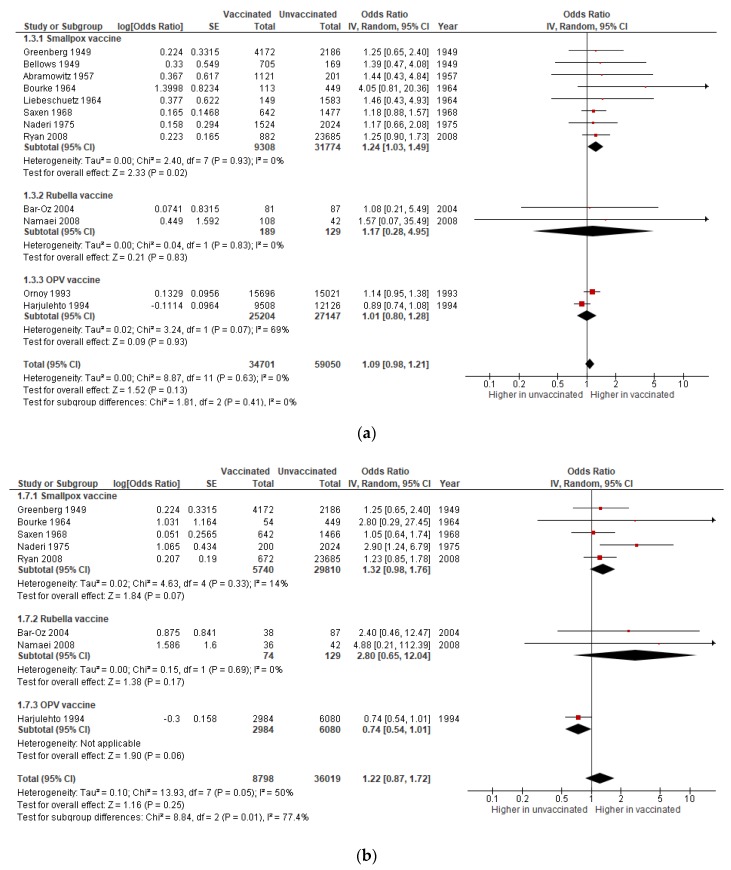
Meta-analysis of the association of maternal immunization with congenital anomalies. Forest plots showing the effect of immunization during pregnancy on the odds of congenital anomalies in all trimesters (**a**) and in the first trimester (**b**), subgrouped by vaccine.

**Table 1 vaccines-08-00124-t001:** Characteristics of included studies.

Study	Location	Study Design	Exposure	Participants	Exposure in 1st Trimester	Previous Immunity	Control Group	Measured Outcomes
Abramowitz et al. (1957) [21]	Cape Town, South Africa	Retrospective cohort	Smallpox vaccine before 20 weeks gestation	1121 vaccinated women (510 with successful vaccination *)	NR	NR	201 women not vaccinated during pregnancy	Stillbirth, birth defects, neonatal death. Outcome definitions not reported
Bar-Oz et al. (2004) [22]	Toronto, Canada	Prospective cohort	Rubella (RA 27/3) vaccination ≤3 months before/after conception	94 women counselled about safety of rubella vaccination during pregnancy through a telephone service	38 women	NR	95 women counselled at similar gestational ages for non-teratogenic exposures, not vaccinated during pregnancy	Miscarriage, birth defects, congenital rubella syndrome, neonatal death. Outcomes reported by mother and physician ≥6 months after expected DOB
Bellows et al. (1949) [23]	New York, U.S.	Prospective cohort	Smallpox vaccination in pregnancy during a mass vaccination campaign	720 vaccinated women (571 successful vaccination *) ≤4 months pregnant at admission to antenatal clinic	247 women	210/720 with accelerated reaction, suggestive of partial immunity	173 women admitted to the same antenatal clinics before 4 months gestation, not vaccinated during pregnancy	Miscarriage (before 5 months), stillbirth (after 5 months), birth defects (physical exam, fundoscopy, X-ray, follow-up for 12 months), neonatal death
Bourke et al. (1964) [24]	Dublin, Ireland	Prospective cohort	Successful smallpox vaccination * at any stage of pregnancy	112 vaccinated women attending antenatal clinics in 4 hospitals that account for >80% of births in Dublin	54 women	NR	448 women attending the antenatal clinics on the same day (4 adjacent charts), not vaccinated during pregnancy	Miscarriage, stillbirth, birth defects (including stillbirths), neonatal death. Outcome ascertained from medical records. Definitions not reported
Ebbin et al. (1973) [25]	Los Angeles County, U.S.	Retrospective cohort	Rubella vaccination during pregnancy or within 3 months before conception	60 vaccinated women admitted to 7 participating hospitals or referred from private physicians	27 women	9/60 previously susceptible, rest unknown	47 controls from hospital or private practice, matched for age, race, parity, sex of the infant, and private/non-private hospital status	Miscarriage, congenital infection (viral isolation in products of conception from abortion cases, or in pharyngeal and rectal swabs from live born infants)
Greenberg et al. (1949) [26]	New York City, U.S.	Retrospective cohort	Smallpox vaccination in 1st trimester of pregnancy during a mass campaign	4172 † infants born to vaccinated women in participating hospitals and health stations	4172 infants	NR	2186 infants born to non-vaccinated women in the same period, identified in participating hospitals and health stations	Birth defects (excluding club foot, hydrocele, inguinal hernia and haemagiomas), LBW
Harjulehto et al. (1993, 1994 and 1995) [27,28,29]	Greater Helsinki Region, Finland	Retrospective cohort (population-based)	OPV vaccination during mass campaign; exposure not determined at the individual level	9508 ‡ births in the 3 hospitals that serve the region, born of women pregnant during the mass vaccination campaign	2984 § births	Most women likely immune (IPV included in national schedule)	12,126 live and stillbirths reported in the same hospitals from July to December 1984 and 1986	Stillbirth (after 22 weeks EGA), prematurity, SGA, birth defects (BPA criteria, including autopsies), neonatal death (in the first 7 days of life)
Liebeschuetz et al. (1964) [30]	London, UK	Retrospective cohort	Successful smallpox vaccination * in pregnancy during a mass campaign	157 vaccinated women attending a maternity hospital during a 6-month period after mass campaign	131 women	105/157 had received smallpox vaccine in the past	1657 women attending the hospital during the same period who were not vaccinated or “unsuccessfully” vaccinated*	Miscarriage, stillbirth, birth defects, fetal vaccinia. Outcome definitions not reported
Naderi et al. (1975) [31]	Shiraz, Iran	Prospective cohort	Successful smallpox vaccination * in pregnancy during mass campaign	1542 infants of 1522 women attending university hospital clinics within 10 months after campaign	211 infants	All exposed women received smallpox vaccine in the past	2045 infants of 2014 women not vaccinated during pregnancy and attending the same clinics during the following year	Miscarriage, stillbirth, prematurity, birth defects, clubfoot. Outcome definitions not reported
Namaei et al. (2008) [32]	Birjand, Iran	Prospective cohort	Rubella vaccine ≤3 months before/after conception, during a mass campaign	106 vaccinated women receiving antenatal care and delivering at a university hospital	71 women	Women with previous infection or vaccination were excluded	40 women not vaccinated during pregnancy. No details of recruitment provided	Stillbirth, prematurity, congenital infection (serology in cord or infant blood), congenital rubella syndrome (CDC’s clinical criteria)
Nishioka et al. (1998) [33]	Uberlandia, Brazil	Case control	Yellow fever (17D) vaccination in pregnancy during a mass campaign §	CASES: 39 women attended for miscarriage at university hospital, with LMP ≥15 days before mass campaign	NR	NR	74 women living in the same city who attended the antenatal clinic at the university hospital	Miscarriage (pregnancy loss before 28 weeks EGA)
Ornoy et al. (1990 and 1993) [34,35]	Israel	Historically-controlled cohort (population-based)	OPV vaccination during a mass campaign with 90% coverage; exposure not determined at the individual level	Women attending hospitals in West Jerusalem within 4 months of campaign (miscarriage) or ≤12 months after (birth defects, LBW) ‡ N = 20,926 annual births	NR	Most women likely immune (poliovirus vaccine in national schedule since 1950s)	Women attending the included hospitals during the same period in the previous year, who were not vaccinated during pregnancy N = 20,143 annual births	Miscarriage (in relation to the number of annual births), birth defects (as proportion of annual live births), LBW (<2500 grs birthweight). Outcomes obtained from hospital records
Ryan et al. (2008) [36]	U.S.	Retrospective cohort. (Dept. of Defense databases)	Smallpox vaccination at any stage of pregnancy	882 infants born during 2003-2004 to active-duty military women vaccinated inadvertently during pregnancy	672 women	NR, but probably not previous immunity (routine vaccination stopped in 1972)	23,685 infants born to military women not vaccinated against smallpox; 6853 infants born to active-duty women vaccinated before or after pregnancy	Prematurity (birth before 37 weeks EGA), birth defects (NBDPN definitions used, records up to 12 months of age reviewed). Outcomes defined using ICD-9-CM codes
Saxen et al. (1968) [37]	Finland	Case control	Smallpox vaccination before or during pregnancy in the context of a country-wide campaign	CASES 835 stillbirths and 642 infants with birth defects notified to the National Board of Health	NR	73% of mothers in study group and 77% in control group previously vaccinated	1477 infants born next after stillbirth/malformed infants in the same district	Stillbirth, birth defects Outcome definitions not reported
Skipetrova et al. (2018) [20]	Several countries (mostly Latin America)	Secondary analysis of data from clinical trials	Dengue (CYD-TDV) vaccination during pregnancy or <30 days before LMP (“risk period”)	58 women inadvertently vaccinated during the “risk period” in CYD-TDV clinical trials	Most women vaccinated before or shortly after conception	NR	341 pregnant women vaccinated outside the “risk period”, 30 received placebo during “risk period”, 179 received placebo outside the “risk period”	Miscarriage (pregnancy loss before 20 weeks EGA), stillbirth (fetal death after 20 weeks EGA)

NR = Not reported, NA = Not applicable, DOB = Date of birth, LBW = Low birth weight, EGA = Estimated gestational age, SGA = Small for gestational age, SD = Standard deviation, IPV = Inactivated poliovirus, CDC = Centers for Disease Control and Prevention, LMP = Last menstrual period, BPA = British Pediatric Association, NBDPN = National Birth Defects Prevention Network. * Successful vaccination against smallpox implies development of a vesicle or pustule at the inoculation site. † N = 4238 in exposed group and 2214 in control group for assessment of LBW (longer follow-up to include all viable infants whose mothers were <3 months pregnant at vaccination). ‡ Exposure status (vaccination during pregnancy) was not determined at the individual level. § Mass campaign after dengue outbreak with use of organophosphate fogging.

**Table 2 vaccines-08-00124-t002:** Summary of findings.

**Vaccination during Pregnancy Compared with No Vaccination.**				
**Patient or population:** Pregnant women and their fetuses/infants. **Intervention:** Administration of live vaccines during pregnancy or shortly before conception. **Comparison:** No exposure to live vaccines during pregnancy or shortly before conception.				
**Outcomes**	**Relative Effect (95% CI)**	**Number of Participants (Studies)**	**Quality of the Evidence (GRADE)**	**Comments**
Miscarriage	OR 0·98 (0.87 to 1.10)	17,763 (9 studies)	Very low	Includes data on smallpox (4 studies), rubella (2), OPV (1), dengue (1), and YF (1) vaccines.
Stillbirth	OR 1·04 (0.74 to 1.48)	32,701 (9 studies)	Very low	Includes data on smallpox (6 studies), rubella (1), OPV (1), and dengue (1) vaccines.
Congenital anomalies	OR 1.09 (0.98 to 1.21)	93,751 (12 studies)	Very low	Includes data on smallpox (8 studies), rubella (2), and OPV (2) vaccines.
Preterm birth	OR 0.99 (0.90 to 1.08)	49,995 (5 studies)	Very low	Includes data on smallpox (2 studies), rubella (2,) and OPV (1) vaccines.
Neonatal death	OR 1.06 (0.68 to 1.65)	24,499 (5 studies)	Very low	Includes data on smallpox (3 studies), rubella (1), and OPV (1) vaccines.
Miscarriage after 1st trimester vaccination	OR 2.66|(0.73 to 9.64)	2832 (3 studies)	Very low	Includes data on smallpox (1 study) and rubella (1) vaccines.

GRADE Working Group grades of evidence. High quality: We are very confident that the true effect lies close to that of the estimate of the effect. Moderate quality: We are moderately confident in the effect estimate: The true effect is likely to be close to the estimate of the effect, but there is a possibility that it is substantially different. Low quality: Our confidence in the effect estimate is limited: The true effect may be substantially different from the estimate of the effect. Very low quality: We have very little confidence in the effect estimate: The true effect is likely to be substantially different from the estimate of effect.

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
