# Peer review of "Safety of Administering Live Vaccines during Pregnancy: A Systematic Review and Meta-Analysis of Pregnancy Outcomes"

_vaccines, 2020, doi:10.3390/vaccines8010124_

Round 1

Reviewer 1 Report

In this manuscript by Almudena Laris-González et al., the authors conducted a meta-analysis based on several databases, searching for adverse pregnancy outcomes associated vaccinations using live-attenuated viruses.

Main findings-         

From a total of 2,831 articles, the authors identified 15 studies that met the inclusion criteria: 12 cohort studies, two case-control studies and one secondary data analysis. Of these 15 studies, eight related to smallpox, three to rubella, two to Oral Polio Vaccine, one to dengue and one to Yellow Fever. Meta-analysis searched for miscarriage, stillbirth, congenital anomalies, prematurity and neonatal death. The smallpox vaccine administered in the first pregnancy trimester is associated with an increased risk of miscarriage. (Fig.2B). The rubella vaccine administered in the first pregnancy trimester is associated with an increased risk of congenital anomalies (Fig.4B). As regards preterm birth and neonatal birth, no association with vaccination was found. For all these issues, the quality of evidence was rated as low, due to serious risks of bias in the primary publications (in four studies, there was a critical bias in the selection of participants).

The authors based their review on 15 studies that relate to five different vaccines (smallpox, rubella, OPV, YF and dengue). All the studies that were included contain serious biases (tables S2-S6), and they relate to very different LAVs, including OPV that is given by the oral route.

It would be more sensible to limit the review to only smallpox and rubella, and perhaps to complete and strengthen this review by taking into account some of the studies that, even if excluded here, could be useful in providing upper bound estimates.

It is somewhat surprising that only 15 studies were retained for analysis, including two studies from 1949 and two from the sixties. 

Major remarks  

Given the very different LAVs considered, including one given by the oral route (OPV), it seems inappropriate to provide pooled estimates, and much more meaningful to give specific estimates for each individual LAV.

The authors do not clearly explain on what basis most of the 2882 records were excluded. Lines 88-90 may perhaps be completed, either in the Methods section or in the Results section in the following manner “Narrative reviews (n= ), case series (n= )….. were excluded from the review”.

Figure 2, 3 and 4. The x-scale of the graphs (odds ratio) must show the subdivisions [2-9] of the logarithmic scale, in order (i) to emphasize the fact that the scale is logarithmic and (ii) to allow the reader to estimate the approximate value of the odds ratio. It is indeed very important to see whether this odd ratio is e.g. 2, 4 or 9. 

Figure 2B. There is no rationale in grouping the three studies (two smallpox studies and one rubella study).The same remark applies to Fig. 2A, 3A 4A and 4B: what is the rationale of pooling studies that relate to different LAVs?

Figure 2C is useless, being redundant with (and less informative than) Fig. 2A, of which it is only a different representation with no added value.

Fig 3B and 4C. Same remark as above, these graphs are redundant with Fig. 3A and 4A, respectively. 

Fig. 4B and lines 219-240. Contrary to the author’s interpretation, it seems that Rubella vaccine administered in the first trimester is associated with an increased risk of congenital anomalies, much more so than smallpox vaccine. 

The discussion lacks a synthetic view. It instead presents several separated studies, each with its own biases.

Minor remarks

Line 69. Furthappermore???

Author Response

We thank the reviewer for the comments, and provide the following point-by-point response:

1) It would be more sensible to limit the review to only smallpox and rubella, and perhaps to complete and strengthen this review by taking into account some of the studies that, even if excluded here, could be useful in providing upper bound estimates:

We believe that a review of the yellow fever data is also highly relevant, given the recent outbreaks of YF. Similarly, emerging infections are likely to be caused by other viruses, hence we think, it is important to the field to include as comprehensive a review of the issue as possible. This was recently also highlighted in a WHO meeting on safety of vaccines in pregnancy.

Uncontrolled cohorts were indeed taken into account in section 3.2 in order to increase our ability to detect safety issues. This allowed us to estimate the upper bound for congenital rubella syndrome (line 268). However, they could not be included in the meta-analysis given the lack of a control group.

2) Given the very different LAVs considered, including one given by the oral route (OPV), it seems inappropriate to provide pooled estimates, and much more meaningful to give specific estimates for each individual LAV.

We agree with the reviewer that it is debatable to obtain pooled estimates. However, the subgroup analyses helped us to analyze the heterogeneity between different vaccines, and specific estimates for each LAV are provided in the forest plots for the different outcomes. We believe that the overall meta-analysis does provide useful information that, nonetheless, needs to be interpreted cautiously. We address this in the discussion (in the section that discusses limitations): “each LAV might have a different effect on pregnancy outcomes, which might be obscured by obtaining pooled effects. However, this was addressed by subgroup analyses which revealed an increased odds of miscarriage and congenital anomalies only after smallpox vaccination.” (lines 347-349).

3) It is somewhat surprising that only 15 studies were retained for analysis, including two studies from 1949 and two from the sixties.

Despite a thorough literature search, which was carried out employing a search strategy constructed in collaboration with a medical librarian, only 15 articles fulfilled the inclusion criteria for the meta-analysis (plus 19 uncontrolled cohorts included in the qualitative synthesis).

We consider this to be expected given that live vaccines have been routinely withheld from pregnant women, which explains the scarcity of available evidence.

With regards to the two studies from 1949 and two from the sixties: these studies occurred in the setting of mass vaccination campaigns against smallpox in the U.S. (1949) and United Kingdom (1960s).

4) The authors do not clearly explain on what basis most of the 2882 records were excluded. Lines 88-90 may perhaps be completed, either in the Methods section or in the Results section in the following manner “Narrative reviews (n= ), case series (n= )….. were excluded from the review”.

Two reviewers independently screened 2882 titles and abstracts, and only 121 were considered relevant for the purpose of this review (2757 were excluded based on irrelevance: the majority of them did not address vaccine safety or concerned only animal vaccines or inactivated vaccines).

Among 121 full-text articles assessed for eligibility, 71 were excluded. Reasons for exclusion are summarized in Fig 1 and we have also listed them below:

12 Uncontrolled cohorts and pregnancy registries

13 Case reports and case series

10 Passive surveillance and pharmacovigilance studies

2 Uncontrolled trials in women scheduled for abortion

25 Reports with no primary data (Reviews, editorials or letters)

3 Live vaccine outcomes mixed with inactivated vaccine data

4 Not concerning pregnancy outcomes

2 Multiple publications (articles published in more than one journal)

5) Figure 2, 3 and 4. The x-scale of the graphs (odds ratio) must show the subdivisions [2-9] of the logarithmic scale, in order (i) to emphasize the fact that the scale is logarithmic and (ii) to allow the reader to estimate the approximate value of the odds ratio. It is indeed very important to see whether this odd ratio is e.g. 2, 4 or 9.

Thanks for the remark. We increased the scale of the forest plots within the following caveats: We wanted to keep the same scale on all the forest plots to avoid confusion when comparing forest plots; the scale should allow for all diamond points to be fully visible (not cropped).

6) Figure 2B. There is no rationale in grouping the three studies (two smallpox studies and one rubella study).The same remark applies to Fig. 2A, 3A 4A and 4B: what is the rationale of pooling studies that relate to different LAVs?

This has been answered in a previous comment regarding pooled estimates.

7) Figure 2C is useless, being redundant with (and less informative than) Fig. 2A, of which it is only a different representation with no added value.

We were confused by this comment, as the funnel plot in Figure 2C is intended to address publication bias and therefore provides information that is not conveyed in the forest plot in figure 2A (and vice-versa). Maybe the reviewer referred to two other, more similar plots? However, we agree that figure 2C is not essential, and has been moved to the supplementary material.

8) Fig 3B and 4C. Same remark as above, these graphs are redundant with Fig. 3A and 4A, respectively.

As above (3B and 4C are intended to address publication bias and, thus, not redundant with 3A and 4A, but has been moved to the supplementary material).

9) Fig. 4B and lines 219-240. Contrary to the author’s interpretation, it seems that Rubella vaccine administered in the first trimester is associated with an increased risk of congenital anomalies, much more so than smallpox vaccine.

Even though the OR for this association is 2.8, the confidence interval is wide and crosses the unity (0.65-12) so we consider there is no evidence for an increased risk of congenital anomalies, although it cannot be ruled out and we have stated this clearly in the discussion

We have added the following explanation to section 3.1.3: “the subgroup analysis revealed an increase in the odds of congenital anomalies after smallpox vaccination (OR: 1·24; 95% CI 1·03-1·49) and a tendency towards an association with rubella vaccine, albeit with a very wide confidence interval (OR: 2.8; 95% CI 0.65-12.04).”

10) The discussion lacks a synthetic view. It instead presents several separated studies, each with its own biases.

We considered important to first discuss the available evidence for each individual vaccine, given the inherent differences between them, and then to proceed to the synthetic view (limitations, implications for future research and decision making).

However, we have added the following remarks in order to broaden our discussion with a more general scope:

“Furthermore, other factors need to be taken into account when considering maternal immunization, including implementation challenges and the need for integration with existing prenatal care services, as well as the potential interference of maternal antibodies with the development of infant humoral immune responses to vaccines in the first few months of life. This interference has been observed for several vaccines but its clinical significance is still uncertain and might differ depending on the specific antigen [77,78].”

“The views of pregnant women themselves need to be taken into account in research design and policy making. Qualitative research on vaccine confidence 42,98 and willingness to participate in clinical trials, conducted in different settings and geographic locations, can help improve acceptance among pregnant women.”

In the context of known increased risks of severe manifestations in pregnant women and/or their infants associated with emerging infections, e.g. Ebola, Zika and Lassa, intense discussions about risks and benefits of the use of novel vaccines in pregnancy are currently ongoing. We hope that our review can inform these discussions and decisions.

Reviewer 2 Report

In the report by Laris-Gonzalez et. al. entitled "Safety of administering live vaccines during pregnancy: a systematic review and meta-analysis of pregnancy outcomes"  the authors provide a fairly comprehensive review of maternal immunization with live-attenuated vaccines (LAV) and the impact on the fetus/infant.  Because of the availability data the authors focused on smallpox, rubella, poliovirus, yellow fever and dengue vaccines.  No association was found between LAV miscarriage, stillbirths, malformations and prematurity.  These findings are a welcome find and should be reported to the scientific community since in the case of Zika, Ebola and newly emerging 2019 nCoV much effort is being placed on mass vaccination strategies.  Predictive information regarding this very unique cohort is therefore very instructive and useful.  

There were a few minor weaknesses to the report the authors should address. While figure 1 and the tables were fine the later figures (2-4) were off low quality and seemed cut and pasted in.  They were also fairly difficult to digest.  I would suggest include the salient data points into the figure and move all the other extra details into either an accompanying table or supplemental figure.  Additionally, a question that emerged that was not really addressed was vaccine formulations between the LAVs and ones in development and the impact.  For example, are adjuvants included and there potential impact.

minor spell check:

p. 2 line 69- "Furthappermore" I believe should be "Furthermore"    

Author Response

We thank the reviewer for these positive comments. The figures in the word file are indeed cut and pasted but the original figures, of good quality, have been submitted in a separate zip file (as .eps documents). In order to simplify figures, we have moved the funnel plots to the supplementary material (mentioning in the text that there was no sign of publication bias) and maintained the forest plots.

As the reviewer points out, other vaccine components were not addressed in this review and could indeed, have an impact in the effect of newer vaccines. We have added the following remark to the discussion, acknowledging this limitation: “The use of modern technologies, such as viral vectored vaccines and novel adjuvants, might also modify the safety profile of live vaccines available in the near future.”  

Reviewer 3 Report

In this well-written and well-organized communication the authors have addressed an important issue in global maternal health - the safety of live vaccine administration to pregnant women. In order to determine this the authors have utilized a meta-analysis format to critically examine the published literature in this area. Their initial search criteria from a variety of databases identified 2,831 articles which were reduced to 15 by employing additional inclusion criteria. The authors then examined each of these articles utilizing the appropriate statistical software and evaluated the association of the vaccine use with such obstetrical outcomes co-variables as miscarriage, stillbirth, neonatal death, congenital infection and fetal malformations. The authors critically discuss their findings, the potential bias in the data, and the importance of vaccine administration in current outbreaks.

This communication is a potentially highly significant contribution to the public health, biomedicine, vaccinology and epidemiology literature that provides the first rigorous examination of the overall safety of live vaccine use during pregnancy. It is well-written and organized, thorough, and highly readable. The statistical analyses are appropriate to the study design and directly addresses the questions posed by the authors. It is well-illustrated and the figures complement the text. The references are inclusive and up-to-date. The figures are well-designed and significantly aid in the understanding of the methodology and results.

I have no suggestions to put forward that would improve the clarity, scientific content or quality of analysis of this valuable communication.

Author Response

We thank the reviewer for these positive comments. We hope to make a useful contribution to the global health field.